# Robust Learning with Implicit Residual Networks

**Viktor Reshniak** [1,*,†] and **Clayton G. Webster** [2,3,†]

1   Data Analysis and Machine Learning, Oak Ridge National Laboratory, Oak Ridge, TN 37831, USA
2   Department of Mathematics, University of Tennessee at Knoxville,
    Knoxville, TN 37996, USA; cwebst13@utk.edu
3   Behavioral Reinforcement Learning Lab (BReLL), Lirio LLC, Knoxville, TN 37923, USA; cwebster@lirio.co
*   Correspondence: reshniakv@ornl.gov
†   These authors contributed equally to this work.

**Abstract:** In this effort, we propose a new deep architecture utilizing residual blocks inspired by implicit discretization schemes. As opposed to the standard feed-forward networks, the outputs of the proposed implicit residual blocks are defined as the fixed points of the appropriately chosen nonlinear transformations. We show that this choice leads to the improved stability of both forward and backward propagations, has a favorable impact on the generalization power, and allows for control the robustness of the network with only a few hyperparameters. In addition, the proposed reformulation of ResNet does not introduce new parameters and can potentially lead to a reduction in the number of required layers due to improved forward stability. Finally, we derive the memory-efficient training algorithm, propose a stochastic regularization technique, and provide numerical results in support of our findings.

**Keywords:** ResNet; stability; robust

## 1. Introduction and Related Works

A large volume of empirical results has been collected in recent years illustrating the striking success of deep neural networks (DNNs) in approximating complicated maps by a mere composition of relatively simple functions [1]. Universal approximation property of DNNs with a relatively small number of parameters has also been shown for a large class of functions [2,3]. The training of deep networks nevertheless remains a notoriously difficult task due to the issues of exploding and vanishing gradients, which become more apparent and noticeable with increasing depth [4]. These issues accelerated efforts of the research community in an attempt to explain this behavior and gain new insights into the design of better architectures and faster algorithms. A promising approach in this direction was obtained by casting evolution of the hidden states $y_t \in Y_t$ of a DNN as a dynamical system [5], i.e.,

$$y_{t+1} = \Phi_t(\gamma_t, y_t), \quad t = 0, ..., T-1,$$

where for each layer $t$, $\Phi_t : \Gamma_t \times Y_t \to Y_{t+1}$ is a nonlinear transformation parameterized by the weights $\gamma_t \in \Gamma_t$, and $Y_t$, $\Gamma_t$ are the appropriately chosen spaces. In the case of a very deep network, when $T \to \infty$, it is convenient to consider the continuous time limit of the above expression such that

$$y(t) = \Phi(\gamma(t), x), \quad t > 0,$$

where the parametric evolution function $\Phi : \Gamma \times Y \to Y$ defines a continuous flow through the input data $y(0) = x \in Y$. Parameter estimation for such continuous evolution can be viewed as an optimal controlling problem [6], given by

$$\min_{\gamma(t)} \mathbb{E}_\mu \left[ L\big(y(T), f(x)\big) + \int_0^T R\big(\gamma(t), y(t)\big) dt \right], \tag{1}$$

$$\text{subject to} \quad y(t) = \Phi\big(\gamma(t), x\big), \tag{2}$$

where $L\big(y(T), f(x)\big)$ is a terminal loss function, $R\big(\gamma(t), y(t)\big)$ is a regularizer, and $\mu$ is a probability distribution of the input–target data pairs $(x, f(x))$. More general models additionally consider continuity in the "patial" dimension as well by using differential [7] or integral formulations [8]. A continuous time formulation based on ordinary differential equations (ODEs) was proposed in [9] with the state Equation (2) of the form

$$\dot{y}(t) = \Phi\big(\gamma(t), y(t)\big). \tag{3}$$

In the work [9], the authors relied on the black-box ODE solvers and used adjoint sensitivity analysis to derive equations for the backpropagation of errors through the continuous system.

The authors of [10] concentrated on the well-posedness of the learning problem for ODE-constrained control and emphasized the importance of stability in the design of deep architectures. For instance, the solution of a homogeneous linear ODE with constant coefficients

$$\dot{y}(t) = Ay(t)$$

is given by

$$y(t) = Qe^{\Lambda t} Q^T x,$$

where $A = Q\Lambda Q^T$ is the eigendecomposition of a matrix $A$, and $\Lambda$ is the diagonal matrix with the corresponding eigenvalues. The similar equation holds for the backpropagation of gradients. To guarantee the efficient propagation of information through the network, one must ensure that the elements of $e^{\Lambda t}$ have magnitudes close to one. This condition, of course, is satisfied when all eigenvalues of the matrix $A$ are imaginary with real parts close to zero. In order to preserve this property, the authors of [10] proposed several time continuous architectures of the form

$$\begin{cases} \dot{y}(t) = \Phi_1\big(\gamma_1(t), y(t), z(t)\big), \\ \dot{z}(t) = \Phi_2\big(\gamma_2(t), y(t), z(t)\big). \end{cases} \tag{4}$$

When $\Phi_1(y, z) = \nabla_z H(y, z)$, $\Phi_2(y, z) = -\nabla_y H(y, z)$, the equations above provide an example of a conservative Hamiltonian system with the total energy $H$.

In the discrete setting of the ordinary feed forward networks, the necessary conditions for the optimal solution of (1) and (2) recover the well-known equations for the forward propagation (state Equation (2)), backward gradient propagation (co-state equation), and the optimality condition, to compute the weights (gradient descent algorithm); see, e.g, [11]. The continuous setting offers additional flexibility in the construction of discrete networks with the desired properties and efficient learning algorithms. Classical feed forward networks (Figure 1, left) is just the particular and the simplest example of such discretization, which is prone to all the issues of deep learning. In order to facilitate the training process, a skip-connection is often added to the network (Figure 1, middle) yielding

$$y = x + h \cdot \Phi(\gamma, x), \tag{5}$$

where $h$ is a positive hyperparameter. Equation (5) can be viewed as a forward Euler scheme to solve the ODE in (3) numerically on the time grid with step size $h$. While it was shown that such residual layers help to mitigate the problem of vanishing gradients and speed-up the training process [12], the scheme has very restrictive stability properties [13]. This can result in the uncontrolled accumulation of errors at the inference stage reducing the generalization ability of the trained network. Moreover, the Euler scheme is not capable of preserving geometric structure of conservative flows and is thus a bad choice for the long time integration of such ODEs [14].

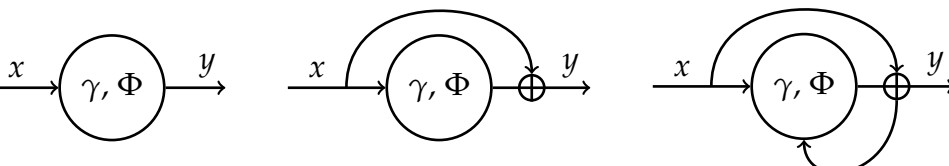

**Figure 1.** From left to right: feed forward layer, residual layer, proposed implicit residual layer.

Memory efficient explicit reversible architectures can be obtained by considering time discretization of the partitioned system of ODEs in (4). The reversibility property allows for recovering the internal states of the system by propagating through the network in both directions and thus does not require one to cache these values for the evaluation of the gradients. First, such architecture (RevNet) was proposed in [15], and, without using a connection to discrete solutions of ODEs, it has the form

$$\begin{cases} y_1 = x_1 + \Phi_1(\gamma, x_2), \\ y_2 = x_2 + \Phi_2(\gamma, y_1), \end{cases} \longleftrightarrow \begin{cases} x_2 = y_2 - \Phi_2(\gamma, y_1), \\ x_1 = y_1 - \Phi_1(\gamma, x_2). \end{cases}$$

It was later recognized as the Verlet method applied to the particular form of the system in (4), see [10,16]. The leapfrog and midpoint networks are two other examples of reversible architectures proposed in [16].

Other residual architectures can be also found in the literature including Resnet in Resnet (RiR) [17], Dense Convolutional Network (DenseNet) [18] and linearly implicit network (IMEXNet) [19]. For some problems, all of these networks show a substantial improvement over the classical ResNet but still have an explicit structure, which has limited robustness to the perturbations of the input data and parameters of the network. Instead, in this effort, we propose new fully implicit residual architecture, which, unlike the above mentioned examples, is unconditionally stable and robust.

As opposed to the standard feed-forward networks, the outputs of the proposed implicit residual blocks are defined as the fixed points of the appropriately chosen nonlinear transformations as follows:

$$y = x + \Phi(\gamma, x, y). \tag{6}$$

The right part of Figure 1 provides a graphical illustration of the proposed layer. One can immediately recognize the feedback loop which is typical for recurrent neural networks (RNN). The standard approach to train RNNs is by backpropagation through time, the algorithm which requires substantial memory resources for deep unrolled recurrent networks. The authors of [20,21] utilized related to our idea of implicit layers to cope with this issue by directly learning the equilibrium points of "infinite" depth recurrent models with a constant memory complexity. The main difference of our approach is that we design the feedback loop with a specific goal of driving the output of the implicit layer to the stable fixed point. We will discuss the choice of the nonlinear transformation *Phi* in (6) and the design of the training algorithm in the next section. It is also worth noting that the idea of learning fixed points is not quite new and is rooting way back to the Hopfield networks and content-addressable ("associative") memory [22,23].

After the first version of this manuscript has appeared, another work proposed the similar idea of implicit Euler skip connections to enhance the adversarial robustness of residual networks [24]. The authors of [24] modified the original residual block by complementing it with a fixed number of steps of the gradient descent algorithm and applied adversarial training to the modified architecture. While both efforts are inspired by implicit numerical schemes for integrating ODEs, our approach is more general as we ensure the convergence of the proposed implicit layer to the **stable** fixed point. As discussed above and unlike the work in [24], in addition to the enhanced stability properties, this approach does not increase the memory complexity of the original residual networks. We also propose an initialization and regularization strategies which allow for training the network efficiently and admit simple interpretation.

The preliminary results for the work proposed in the current manuscript have been presented at the Second Symposium on Machine Learning and Dynamical Systems in the Fields Institute [25]. Here, we provide a completely revised version of this work including an in-depth description of the method, a new regularization approach, and much extended numerical results.

## 2. Description of the Method

We first motivate the necessity for our new method by letting the continuous model of a network be given by the ordinary differential equations in (4) that is:

$$\begin{cases} \dot{y}(t) = \Phi_1\big(\gamma_1(t), y(t), z(t)\big), \\ \dot{z}(t) = \Phi_2\big(\gamma_2(t), y(t), z(t)\big). \end{cases}$$

An s-stage Runge–Kutta (RK) method for the approximate solution of the above equations is given by

$$k_i = \Phi_1\left(\gamma_1(t_0 + c_i h), y_0 + h\sum_{j=1}^{s} a_{ij}k_j, z_0 + h\sum_{j=1}^{s} \hat{a}_{ij}l_j\right)$$

$$l_i = \Phi_2\left(\gamma_2(t_0 + \hat{c}_i h), y_0 + h\sum_{j=1}^{s} a_{ij}k_j, z_0 + h\sum_{j=1}^{s} \hat{a}_{ij}l_j\right) \qquad i, j = 1, .., s \qquad (7)$$

$$y_1 = y_0 + h\sum_{i=1}^{s} b_i k_i, \qquad z_1 = z_0 + h\sum_{i=1}^{s} \hat{b}_i l_i.$$

The order conditions for the coefficients $a_{ij}$, $b_i$, $c_i$, $\hat{a}_{ij}$, $\hat{b}_i$, and $\hat{c}_i$, which guarantee that convergence of the numerical solution is well known and can be found in any topical text, see, e.g., [13]. Note that, when $a_{ij} \neq 0$ or $\hat{a}_{ij} \neq 0$ for at least some $j \geq i$, the scheme is implicit and a system of nonlinear equations has to be solved at each iteration which obviously increases the complexity of the solver. Nevertheless, the following example illustrates the benefits of using implicit approximations.

### 2.1. Linear Stability Analysis

Consider the following linear differential system:

$$\dot{y}(t) = -\omega^2 z(t), \qquad \dot{z}(t) = y(t) \qquad (8)$$

and four simple discretization schemes [13]:

Forward Euler: $\quad y_1 = y_0 - h\omega^2 z_0, \quad z_1 = z_0 + hy_0,$

Backward Euler: $\quad y_1 = y_0 - h\omega^2 z_1, \quad z_1 = z_0 + hy_1,$

Trapezoidal: $\quad y_1 = y_0 - \dfrac{h\omega^2}{2}(z_0 + z_1), \quad z_1 = z_0 + \dfrac{h}{2}(y_0 + y_1),$

Verlet: $\quad y_{0.5} = y_0 - \dfrac{h\omega^2}{2}z_0, \quad z_1 = z_0 + hy_{0.5}, \quad y_1 = y_{0.5} - \dfrac{h\omega^2}{2}z_1.$

Due to linearity of the system in (8), the numerical solution after $n$ steps can be written as

$$\begin{pmatrix} y_n \\ z_n \end{pmatrix} = R^n(h\omega)\begin{pmatrix} y_0 \\ z_0 \end{pmatrix}. \tag{9}$$

The long time behavior of the discrete dynamics is hence determined by the spectral radius of the matrix $R(h\omega)$ (called the *stability matrix*), which needs to be less than or equal to one for the sake of stability. For example, we have $\lambda_{1,2} = 1 \pm ih\omega$ for the forward Euler scheme, and the method is unconditionally unstable. The Backward Euler scheme gives $\lambda_{1,2} = (1 \pm ih\omega)^{-1}$ and the method is unconditionally stable. The corresponding eigenvalues of the trapezoidal scheme have a magnitude equal to one for all $\omega$ and $h$. Finally, the characteristic polynomial for the matrix of the Verlet scheme is given by $\lambda^2 - (2 - h^2\omega^2)\lambda + 1$, i.e., the method is only conditionally stable when $|h\omega| \le 2$.

Figure 2 illustrates this behavior for the particular case of $\omega = 50$. Notice that the flows of the forward and backward Euler schemes are strictly expanding and contracting; if one had to fit the exact flow of the system in (8) using either of these methods, this would result in an inherently ill-posed problem. On the contrary, the implicit trapezoidal and explicit Verlet schemes seem to reproduce the original flow very well, but the latter is conditional on the size of the step $h$. Another nice property of the trapezoidal and Verlet schemes is their symmetry with respect to the exchanging $y_n \leftrightarrow y_{n-1}$ and $z_n \leftrightarrow z_{n-1}$. Such methods play a central role in the geometric integration of reversible differential flows and are handy in the construction of the memory efficient reversible network architectures. Conditions for the reversibility of general Runge–Kutta schemes can be found in [14].

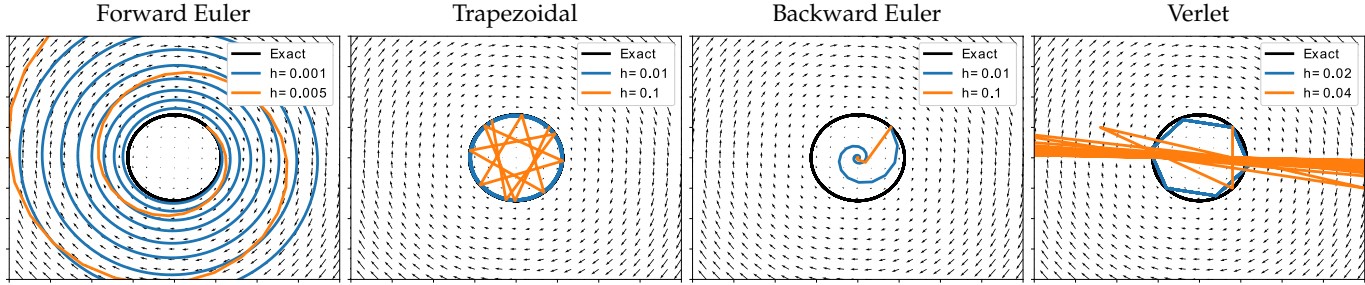

**Figure 2.** Phase diagrams of different numerical solutions of the system in (8).

The discussion above highlights the importance of the appropriate choice for both the structure of a dynamical system (DS) and the corresponding time integrator. The stability of a general Runge–Kutta method is often studied in application to the simpler scalar test equation

$$\dot{y}(t) = \lambda y, \tag{10}$$

and by analogy with (9) its stability function is given by

$$\mathfrak{R}(z) = \frac{\det\left(I - zA + z\mathbb{1}b^T\right)}{\det\left(I - zA\right)}, \qquad z = h\lambda,$$

where $\mathbb{1} = (1, ..., 1)^T$ and $A = (a_{ij})_{i,j=1}^s$, $b = (b_j)_{j=1}^s$ are the parameters of the RK method in (7) with the remaining parameters set to zero for the scalar test equation. From this expression, one can see that the stability function of **any** explicit Runge–Kutta method is a polynomial and hence is bounded only for $z$ in a finite region of the complex plane. The set of all such $z$ with $R(z) < 1$ is called the *stability region* of the method.

### 2.2. Implicit ResNet

Motivated by the discussion above, we propose an implicit residual layer in (6) with a nonlinear map $\Phi(\gamma, x, y)$ given by

$$\Phi(\gamma, x, y) := (1 - \theta)F(\gamma, x) + \theta F(\gamma, y), \qquad \theta \in [0, 1], \tag{11}$$

$$\text{or} \quad \Phi(\gamma, x, y) := F(\gamma, (1 - \theta)x + \theta y), \tag{12}$$

where $x, y, \gamma$ are the input, output, and parameters of the layer, and $F(\gamma, x)$ is a vector field to be estimated. Table 1 shows the derivatives of the nonlinear maps in (11) and (12) with respect to their arguments.

The stability function of the layers in (11) and (12) is given by

$$\mathfrak{R}(z) = \frac{1 + (1 - \theta)z}{1 - \theta z}. \tag{13}$$

The corresponding stability regions are illustrated in Figure 3 indicating the improved stability of implicit layers for increasing $\theta$.

$\theta = 0.00 \qquad \theta = 0.25 \qquad \theta = 0.50 \qquad \theta = 0.75 \qquad \theta = 1.00$

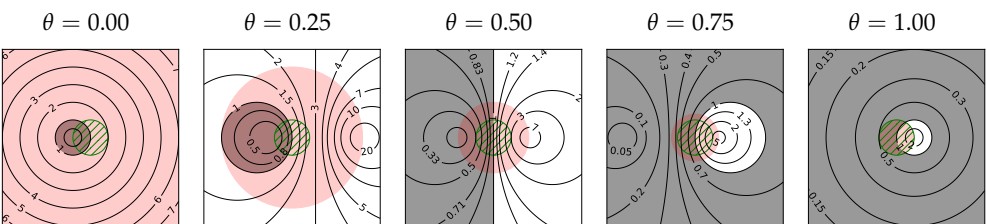

**Figure 3.** Stability regions (grey) and the contours of the stability function of implicit residual layers; the corresponding regions where layers are contractive are shown in red while the hatched circles contain the spectrum of the spectrally normalized 1-Lipschitz function $F(\gamma, x)$.

**Table 1.** Derivatives of the nonlinear maps in (11) and (12).

| $\Phi(\gamma, x, y)$ | $(1 - \theta)F(\gamma, x) + \theta F(\gamma, y)$ | $F(\gamma, z), \quad z = (1 - \theta)x + \theta y$ |
|---|---|---|
| $\frac{\partial \Phi(\gamma, x, y)}{\partial x}$ | $(1 - \theta)\frac{\partial F(\gamma, x)}{\partial x}$ | $(1 - \theta)\frac{\partial F(\gamma, z)}{\partial z}$ |
| $\frac{\partial \Phi(\gamma, x, y)}{\partial y}$ | $\theta\frac{\partial F(\gamma, y)}{\partial y}$ | $\theta\frac{\partial F(\gamma, z)}{\partial z}$ |
| $\frac{\partial \Phi(\gamma, x, y)}{\partial \gamma}$ | $(1 - \theta)\frac{\partial F(\gamma, x)}{\partial \gamma} + \theta\frac{\partial F(\gamma, y)}{\partial \gamma}$ | $\frac{\partial F(\gamma, z)}{\partial \gamma}$ |

Additionally, instead of a single layer in (6), one might consider a block of implicit layers on a given "time interval" $t \in [0, T]$

$$y_t = y_{t-1} + \Phi(\gamma_t, y_{t-1}, y_t), \qquad t = 1, \ldots, T, \tag{14}$$
$$y_0 = x.$$

Note that, by viewing (14) as a numerical ODE integrator and to get the above expressions, we have assumed that the ODE coefficients $\gamma(t)$ are piecewise constant cádlág functions, see Figure 4.

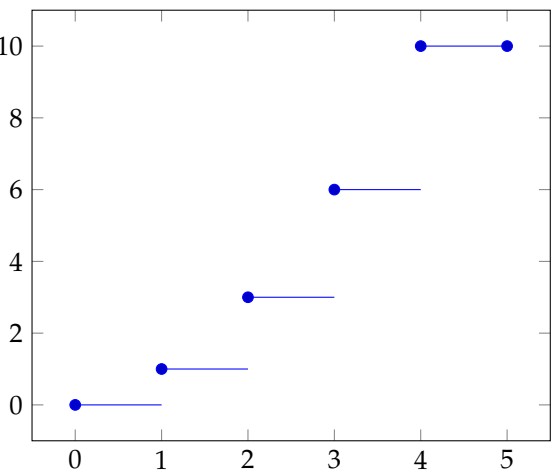

**Figure 4.** Cádlág function.

We will use (12) as the definition of implicit layers in the rest of this work.

### 2.2.1. Forward Propagation

Assume that the fixed point in (6) exists and can be computed. To solve the corresponding nonlinear equation, consider the equivalent minimization problem

$$\min_y \|r(y)\|^2, \qquad r(y) := y - x - \Phi(\gamma, x, y). \tag{15}$$

One way to construct the required solution is by applying the descent algorithm

$$y^{k+1} \leftarrow y^k - \lambda^k s^k, \quad k = 0, 1, 2, \dots$$

where $s^k$ is the descent direction and $\lambda^k$ is the corresponding step size. Several common choices are summarized in Table 2.

**Table 2.** Common descent algorithms [26].

| Method | $s^k$ | $\lambda^k$ | Parameters |
|---|---|---|---|
| gradient descent | $\Lambda^k \cdot \nabla_y \|r(y^k)\|^2 +$ $m^k/\lambda^k$ | Wolfe conditions | scaling matrix $\Lambda^k$ momentum vector $m^k$ |
| quasi-Newton | $(H^k)^{-1} \cdot \nabla_y \|r(y^k)\|^2$ | Wolfe conditions | approximate Hessian $H^k$ |
| conjugate gradient | $\nabla_y \|r(y^k)\|^2 - \beta^k s^{k-1}$ | $\arg\min_\lambda$ $\|r(y^k + \lambda s^k)\|^2$ | conjugate direction parameters $\beta^k$ |

The required gradient of the residual norm can be computed as

$$\nabla_y \|r(y)\|^2 = 2 \cdot \frac{\partial r(y)}{\partial y}^T \cdot r(y), \tag{16}$$

where the Jacobian of the residual vector is given by (c.f. Table 1)

$$\frac{\partial r(y)}{\partial y} = I - \frac{\partial \Phi(\gamma, x, y)}{\partial y}.$$

The expression in (16) can be efficiently computed with the automatic differentiation capabilities of any standard deep learning framework making it possible to interface existing iterative solvers. However, optimized implementations of the gradient descent algorithms are readily provided by any such framework. Hence, the forward propagation for implicit layers can be easily implemented using built-in tools native to each framework and without interfacing external solvers. Listing 1 shows the PyTorch pseudocode of the implicit layer and the left part of Figure 5 illustrates its corresponding computational graph.

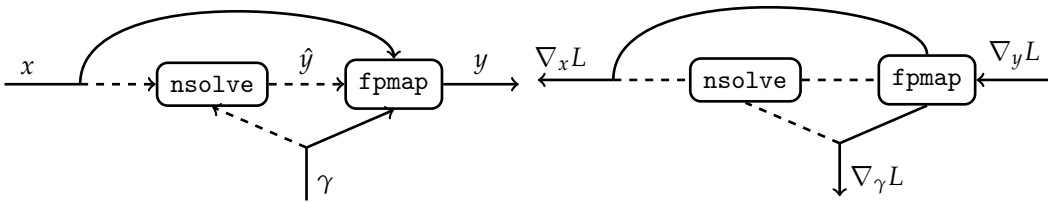

**Figure 5.** Forward and backward computational graphs of the implicit layer.

**Listing 1.** PyTorch pseudocode of the implicit residual block.

```
import torch

class fpmap(torch.Function):
@staticmethod
def forward(ctx,γ,x,ŷ):
y = x + Φ(γ,x,ŷ)
ctx.save_for_backward(y,ŷ)
return y

@staticmethod
def backward(ctx,∇ᵧL):
y,ŷ = ctx.saved_tensors
return (I − ∂y/∂ŷ)⁻ᵀ∇ᵧL

def ImplicitResidualBlock(x):

nsolve = lambda γ,x: arg min_z ‖z − fpmap(γ,x,z)‖²

ŷ = nsolve(γ.detach(),x.detach())

return fpmap.apply(γ,x,ŷ)
```

### 2.2.2. Backpropagation

We now show that, even though the nonlinearity in (6) adds to the complexity of the forward propagation, the direct backpropagation through the nonlinear solver is not required. Firstly, using the chain rule, we can easily find the Jacobian matrices of the implicit residual layer as follows:

$$
\frac{\partial y}{\partial x} = I + \frac{\partial \Phi(\gamma, x, y)}{\partial x} + \frac{\partial \Phi(\gamma, x, y)}{\partial y}\frac{\partial y}{\partial x} = \left(I - \frac{\partial \Phi(\gamma, x, y)}{\partial y}\right)^{-1}\left(I + \frac{\partial \Phi(\gamma, x, y)}{\partial x}\right),
$$

and

$$
\frac{\partial y}{\partial \gamma} = \frac{\partial \Phi(\gamma, x, y)}{\partial \gamma} + \frac{\partial \Phi(\gamma, x, y)}{\partial y}\frac{\partial y}{\partial \gamma} = \left(I - \frac{\partial \Phi(\gamma, x, y)}{\partial y}\right)^{-1}\frac{\partial \Phi(\gamma, x, y)}{\partial \gamma}.
$$

The backpropagation formulas then follow immediately:

$$\nabla_x L = \left( I + \frac{\partial \Phi(\gamma, x, y)}{\partial x} \right)^T \overline{\nabla_y L} \qquad \text{and} \qquad \nabla_\gamma L = \frac{\partial \Phi(\gamma, x, y)}{\partial \gamma}^T \overline{\nabla_y L},$$

where $\overline{\nabla_y L}$ is the solution to the linear system

$$\left( I - \frac{\partial \Phi(\gamma, x, y)}{\partial y} \right)^T \overline{\nabla_y L} = \nabla_y L. \tag{17}$$

Note that the custom backpropagation for the `fpmap` function in Listing 1 is responsible for the linear solve in (17). The gradients of the loss with respect to the parameters and the input are then computed automatically by the deep learning framework. Finally, DL frameworks allow for the cheap computation of the vector-Jacobian products in (17) and hence for the efficient implementation of iterative linear solvers. For example, we utilize restarted GMRES as a linear solver in our implementation.

### 2.3. fpResNet

Sophisticated solvers are not required for the nonlinear and linear systems in (15) and (17) when $\Phi(\gamma, x, y)$ is a contractive mapping, i.e., when

$$Lip(\Phi) := \sup_y \left\| \frac{\partial \Phi(\gamma, x, y)}{\partial y} \right\|_2 = \theta \sup_y \left\| \frac{\partial F(\gamma, y)}{\partial y} \right\|_2 < 1, \tag{18}$$

where $\| \cdot \|_2$ is the spectral norm of the matrix equal to its largest singular value. The red color in Figure 3 is used to highlight the part of the complex plane $\{z : |z| < \theta^{-1}\}$, where the above condition is satisfied for the Dahlquist test equation in (10). In this case, the Banach fixed-point theorem ensures the convergence of the recurrence relation

$$y^{k+1} = x + \Phi(\gamma, x, y^k), \qquad k = 0, 1, \ldots$$

during the forward propagation. The same condition guarantees the validity of the Neumann series expansion for the matrix inverse required for the backpropagation; one gets

$$\overline{\nabla_y L} = \left( I - \frac{\partial \Phi(\gamma, x, y)}{\partial y} \right)^{-T} \nabla_y L = \sum_{i=0}^{\infty} \left( \frac{\partial \Phi(\gamma, x, y)}{\partial y}^T \right)^i \nabla_y L = \sum_{i=0}^{\infty} \nabla_y^i L,$$

with

$$\nabla_y^i L = \frac{\partial \Phi(\gamma, x, y)}{\partial y}^T \nabla_y^{i-1} L, \qquad \nabla_y^0 L = \nabla_y L.$$

Similarly to (17), each $\nabla_y^i L$ has a form of the vector-Jacobian product which can be efficiently evaluated with any deep learning framework. In practice, however, such simple iterations converge linearly with the rate proportional to the Lipschitz constant $Lip(\Phi)$ which can be rather inefficient when $Lip(\Phi) \approx 1$.

### 2.4. Regularization

One way to ensure the stability of the neural network when viewed as a nonlinear DS is by imposing a hard constraint on its parameters to make it globally dissipative or conservative. This approach has been utilized, for instance, in [7,10] and applied to several explicit, and hence only conditionally stable, residual network architectures. Another approach that we employ here is by imposing the structure by regularization. In this section, we review some common regularization techniques and propose a new one that suits the presented implicit architecture.

2.4.1. Lipschitz Continuous Architectures

Enforcing Lipschitz continuity of neural networks has been recognized as an important component in many applications. For instance, explicit bounds on the Lipschitz constants of the loss function have been utilized to improve the robustness and establish the generalization error of large margin classifiers and GAN discriminators in [27–30], respectively. Lipschitz continuity has also been considered implicitly in [31,32] to improve the adversarial robustness of DNNs and in [33] for the design of contractive auto-encoders. In all these works, bounding the Lipschitz constants was implemented by penalizing the norm of the Jacobian matrix of the network on the training dataset.

Spectral normalization of weight matrices can be used to provide the **uniform** bound on the Lipschitz constant of the network. It has been applied to improve the stability of deep networks in [34–36] and to construct invertible normalizing flows of probability distributions in [37]. To justify the method, consider the common choice of $F(\gamma, x)$ in (11) as a composition of affine maps and contractive nonlinear activations, i.e.,

$$F(\gamma, x) = \phi_n \circ \phi_{n-1} \circ ... \circ \phi_1 \circ x \quad \text{with} \quad \phi_i \circ x = \sigma(\gamma_i \circ x + b_i) \quad \text{s.t.} \quad \|\sigma\| \leq 1.$$

The Lipschitz constant of $F(\gamma, x)$ as a function of $x$ can be bounded from above by

$$Lip(F) := \sup_x \left\| \frac{\partial F(\gamma, x)}{\partial x} \right\|_2 \leq \prod_{i=1}^n \|\gamma_i\|_2 = \prod_{i=1}^n \sqrt{\rho(\gamma_i^T \gamma_i)},$$

where $\| \cdot \|_2$ is the spectral norm and $\rho(A)$ denotes the spectral radius of a linear map $A$. Hence, to ensure that $Lip(F) \leq \alpha$, it is enough to take

$$\tilde{\gamma}_i = \frac{\gamma_i}{\max\left(1, \frac{\|\gamma_i\|_2}{\alpha_i}\right)} \quad \text{s.t.} \quad \prod_{i=1}^n \alpha_i \leq \alpha.$$

The exact calculation of the operator norm is expensive and one usually appeals to approximate techniques such as the power iteration method [38]. According to this method, the dominant singular vector $v$ and the singular value $\mu$ of a linear operator $A$ are estimated iteratively as

$$v^{k+1} = \frac{A^T A v^k}{\|A^T A v^k\|}, \qquad \mu^k = \sqrt{(v^k)^T A^T A v^k} = \|A v^k\|.$$

In practice, it is enough to take a fixed number (usually 1) of iterations at each weight evaluation during the training stage since the parameters are not expected to change much close to the convergence of the training loop. By observing that

$$A^T A v^k = \frac{1}{2} \frac{\partial \|A v^k\|^2}{\partial v^k} = \frac{1}{2} \frac{\partial (\mu^k)^2}{\partial v^k},$$

Algorithm 1 provides a simple implementation of this approach.

---

**Algorithm 1** Power iteration method

---

**Input:** linear map $A$, max iterations $k_{max}$
Initialize $v^0$
$v^0 \leftarrow v^0 / \|v^0\|$
**for** $k = 1, ..., k_{max}$ **do**
    $\mu \leftarrow \|Av^k\|$
    $v^{k+1} \leftarrow 0.5 \cdot \partial \mu^2 / \partial v^k$
    $v^{k+1} \leftarrow v^{k+1} / \|v^{k+1}\|$
**end for**
**Output:** $\mu$

---

More generally, spectral normalization allows for controlling the spread of the Jacobian matrix $\dfrac{\partial F(\gamma, x)}{\partial x}$, i.e., the largest distance between its eigenvalues

$$s\left(\frac{\partial F(\gamma, x)}{\partial x}\right) := \max_{i,j}\{|\lambda_i - \lambda_j|\}.$$

By definition of the spectral radius, one has

$$\rho\left(\frac{\partial F(\tilde{\gamma}, x)}{\partial x}\right) \leq \left\|\frac{\partial F(\tilde{\gamma}, x)}{\partial x}\right\|_2 \leq 1 \qquad \rightarrow \qquad Re(\lambda_i) \in [-1, 1] \quad \forall i.$$

Hence, by denoting $F^{-1,1}(\gamma, x) := F(\tilde{\gamma}, x)$, we obtain

$$F^{\alpha,\beta}(\gamma, x) = \frac{\alpha + \beta}{2}x + \frac{\beta - \alpha}{2}F^{-1,1}(\gamma, x) \qquad \rightarrow \qquad Re(\lambda_i) \in [\alpha, \beta] \quad \forall i,$$

so that all eigenvalues of the Jacobian $\dfrac{\partial F^{\alpha,\beta}(\gamma, x)}{\partial x}$ are located in the disc with radius $(\beta - \alpha)/2$ centered at $(\alpha + \beta)/2$. In practice, we use the more flexible form of this definition

$$F^{\alpha,\beta}(\gamma, x) = \frac{\alpha + \beta}{2}x + \frac{\beta - \alpha}{2}S(\vartheta) \odot F^{-1,1}(\gamma, x), \tag{19}$$

where $\odot$ is the Hadamard product, $\vartheta$ are additional learnable parameters, $S(\vartheta)$ has the same dimensionality as $F^{-1,1}(\gamma, x)$, and each $S_i(\vartheta_i) \in (0, 1)$ is the sigmoid function.

2.4.2. Trajectory Regularization

The Lipschitz constant of the proposed residual block in (6) is defined as

$$Lip(y) := \sup_x \left\|\left(I - \frac{\partial \Phi(\gamma, x, y)}{\partial y}\right)^{-1}\left(I + \frac{\partial \Phi(\gamma, x, y)}{\partial x}\right)\right\|_2,$$

and, for the linear scalar test equation in (10), it is given by the stability function (13) of the method. The hatched circle in Figure 3 contains the spectrum of the 1-Lipschitz function $F(\tilde{\gamma}, x)$ and shows that the Lipschitz constant of a residual block can fall outside of its stability region. Moreover, the pole of (13) is located at $z = \theta^{-1}$ and the stability of implicit layers might actually degrade with increasing $\theta \in [0, 1]$ if no additional precautions are taken to isolate the spectrum of the layers from its vicinity. This can be achieved, for instance, by setting $F(\gamma, x) := F^{\alpha, \theta^{-1}}(\gamma, x)$ for some $\alpha < \theta^{-1}$.

Previous works have focused on improving the efficiency of residual and neural ODE architectures by regularizing their vector fields in a way that leads to a simpler dynamical behavior. For example, the authors of [39] considered the following regularizer

$$R(\gamma) := \alpha_K \int_0^T \|F(\gamma(t), y(t))\|^2 dt + \alpha_J \int_0^T \left\| \frac{\partial F(\gamma(t), y(t))}{\partial y} \right\|_F^2 dt.$$

The first term encourages the trajectories with origin in the training dataset to follow straight lines, while the second term reduces overfitting by restricting the vector field to be nearly constant in the vicinity of each trajectory. A more general approach has been taken in [40] by directly penalizing the *K*-th order total derivative of the vector field along the solution trajectories. It has been shown that, by matching *K* to the order of numerical integrator, it is possible to significantly reduce the cost of solving the learned dynamics without sacrificing the resulting accuracy.

Instead, we consider "discrete-time" regularization of the form

$$R(\gamma) := \frac{1}{T} \left( \sum_{t=0}^{T}{}' \left[ \frac{\alpha_{div}}{d} \left( \frac{t}{T} \right)^p \nabla \cdot F(\gamma_t, y_t) + \frac{\alpha_{jac}}{d^2} \left\| \frac{\partial F(\gamma_t, y_t)}{\partial y_t} \right\|_F^2 \right] + \frac{\alpha_{TV}}{T} \sum_{t=1}^{T} \|\gamma_t - \gamma_{t-1}\|^2 \right), \tag{20}$$

where $\sum'$ is the trapezoidal quadrature rule, $d$ is the dimension of the vector field $F : \Gamma \times \mathbb{R}^d \to \mathbb{R}^d$, $p \geq 0$ is some fixed number, and $\nabla \cdot F(\gamma_t, y_t)$ is the divergence of $F(\gamma_t, y_t)$.

The total variation (TV) like term in (20) is responsible for the temporal regularity of the vector field $F(\gamma(t), y(t))$. Similarly to the approach of [39,40], the second term penalizes the Jacobian matrix of the layer producing vector fields which tend to be constant in the vicinity of the learned trajectories; this results in simpler dynamics and faster convergence. Finally, the first term in (20) promotes the negative divergence of $F(\gamma_t, y_t)$ along the trajectories. By definition of the divergence of the vector field, one has

$$\nabla \cdot F(\gamma_t, y_t) = \sum_{i=1}^{d} \left( \frac{\partial F(\gamma_t, y_t)}{\partial y_t} \right)_{ii} = \mathrm{Tr} \left( \frac{\partial F(\gamma_t, y_t)}{\partial y_t} \right) = \sum_{i=1}^{d} \lambda_i \left( \frac{\partial F(\gamma_t, y_t)}{\partial y_t} \right),$$

i.e., it is equal to the sum of eigenvalues of the Jacobian matrix. By minimizing this term, we attempt to push the spectrum of the Jacobian matrix to the negative part of the complex plane so that we can take advantage of the enhanced stability of implicit layers in this region. In addition, note that, by definition, the squared Frobenius norm of a matrix is equal to the sum of its squared singular values

$$\left\| \frac{\partial F(\gamma_t, y_t)}{\partial y_t} \right\|_F^2 = \sum_{i=1}^{d} \sum_{j=1}^{d} \left( \frac{\partial F(\gamma_t, y_t)}{\partial y_t} \right)_{ij}^2 = \mathrm{Tr} \left( \frac{\partial F^T(\gamma_t, y_t)}{\partial y_t} \frac{\partial F(\gamma_t, y_t)}{\partial y_t} \right) = \sum_{i=1}^{d} \sigma_i^2 \left( \frac{\partial F(\gamma_t, y_t)}{\partial y_t} \right).$$

Hence, the first two terms in (20) are competing with each other, and the coefficient $(t/T)^p$ is used to balance these two components by increasing the level of dissipation along the trajectories. The divergence term, however, does not impact the off-diagonal part of the Jacobian matricies, and, for large enough $\alpha_{div}$, this will promote their diagonal dominance.

According to the Gershgorin circle theorem, every eigenvalue of a matrix *A* lies within at least one of the Gershgorin discs $D(a_{ii}, r_i)$ with $r_i = \sum_{i \neq j} |a_{ij}|$. This means that the optimal solution of the optimization problem with the proposed regularizer in (20) will result in the dynamics that are strongly dissipative in the directions irrelevant for the accurate representation of the training data effectively reducing the dimension of the state space. The behavior of the dynamical system in the remaining directions along the so-obtained low-dimensional state manifold can potentially be arbitrary and, with a proper balance between the loss and regularization, should not decrease the expressive power of the network.

For the efficient evaluation of the divergence and Jacobian regularizers in (20), we utilize the unbiased stochastic Hutchinson trace estimator [41] to obtain

$$\nabla \cdot F(\gamma_t, y_t) = \mathbb{E}_{z \sim \mathcal{N}(0,1)} \left[ z^T \frac{\partial F(\gamma_t, y_t)}{\partial y_t} z \right],$$

$$\left\| \frac{\partial F(\gamma_t, y_t)}{\partial y_t} \right\|_F^2 = \mathbb{E}_{z \sim \mathcal{N}(0,1)} \left[ \left\| z^T \frac{\partial F(\gamma_t, y_t)}{\partial y_t} \right\|^2 \right].$$

This approach has also been applied in [39,42]. Algorithm 2 provides a more general variant of Hutchinson algorithm which allows for estimating the diagonal of a matrix [43]. This can be convenient when one needs to control the magnitude of the diagonal elements rather than just their sum.

---

**Algorithm 2** Stochastic diagonal estimator

---

**Input:** linear map $A$, max iterations $k_{max}$
Initialize $D^0 = 0$
**for** $k = 1, ..., k_{max}$ **do**
  $v^k \leftarrow \mathcal{N}(0, 1)$
  $t^k \leftarrow t^{k-1} + \left( A(v^k) \odot v^k \right)$
  $q^k \leftarrow t^{k-1} + \left( v^k \odot v^k \right)$
  $D^k \leftarrow t^k \oslash q^k$
**end for**
**Output:** Approximate diagonal $D^{k_{max}}$ of $A$

---

## 3. Results

The source code used to generate all examples below can be found at https://github.com/vreshniak/ImplicitResNet.

### 3.1. Example 1. (Regression)

For the first example, we consider the simple problem that can be easily visualized. The goal is to approximate the one-dimensional sine function in Figure 6 given $N = 20$ data points evenly distributed on the interval $x \in [-5, 5]$. Following the approach of [44], we augment the original one-dimensional data with an additional dimension initialized with zero. The resulting two-dimensional vector field $F(\gamma, y) : \Gamma \times \mathbb{R}^2 \rightarrow \mathbb{R}^2$ is approximated by a multilayer perceptron using three hidden layers of width 10 and GeLU activation function, i.e.,

$$F(\gamma, y) = \gamma_{out} \sigma \left( \gamma_3 \sigma \left( \gamma_2 \sigma \left( \gamma_1 \sigma (\gamma_{in} y + b_0) + b_1 \right) + b_2 \right) + b_3 \right),$$

where $\gamma_{in} \in \mathbb{R}^{10 \times 2}$, $\gamma_{out} \in \mathbb{R}^{2 \times 10}$, $\gamma_i \in \mathbb{R}^{10 \times 10}$ and $b_i \in \mathbb{R}^{10 \times 1}$.

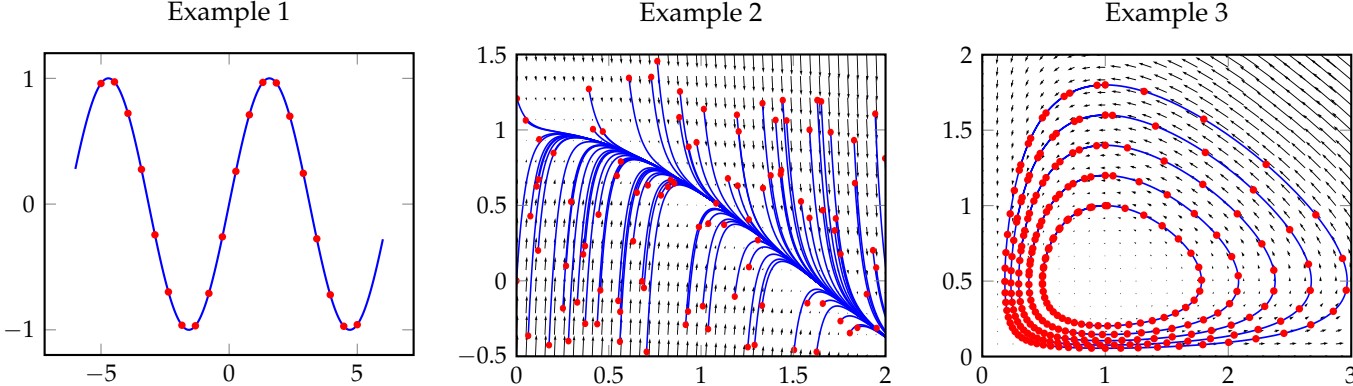

**Figure 6.** Training data in Examples 1–3.

We used $T = 5$ residual layers with shared parameters initialized with a Xavier uniform initializer and trained the network for 3000 epochs using Adam optimizer and the regularized loss given by

$$L(\gamma) := \frac{1}{N} \sum_{i=1}^{N} \left( \left\| g(y_T^i) - f(x^i) \right\|^2 + \frac{1}{T} \sum_{t=0}^{T}{}' \left[ \frac{\alpha_{div}}{d} \left( \frac{t}{T} \right)^2 \nabla \cdot F(\gamma, y_t^i) + \frac{0.1}{d^2} \left\| \frac{\partial F(\gamma, y_t^i)}{\partial y_t^i} \right\|_F^2 \right] \right), \tag{21}$$

where $d = 2$ is the dimension of the hidden state space and $g(y_T)$ gives the last component of $y_T$. The initial learning rate was set to $10^{-3}$ and reduced dynamically using `ReduceLROnPlateau` PyTorch scheduler with `patience` and `cooldown` parameters set to 50 epochs.

Figure 7 shows the learned vector fields and eigenvalues of the Jacobian $\frac{\partial F(\gamma, y)}{\partial y}$ along the learned trajectories for several values of $\theta$ and $\alpha_{div}$. Additionally, the top row of Figure 8 depicts the evolution of the loss components in (21), and the number of nonlinear iterations of the trained network for the selected parameter values. One can see that, with the Jacobian regularization alone ($\alpha_{div} = 0$, $\alpha_{jac} = 0.1$), implicit methods demonstrate similar dynamical behavior for all considered values of $\theta$: (1) the learned vector fields take full advantage of all two available dimensions and tend to be expansive, note the increasing divergence in Figure 8, (2) the learned trajectories mostly follow straight lines, and (3) aside from the fully explicit scheme ($\theta = 0$), the costs of all the methods are nearly identical, see the bottom row of Figure 8. By increasing $\alpha_{div}$, one starts observing the formation of a lower dimensional invariant manifold with increasingly dissipative orthogonal dynamics indicated by the negative part of the spectrum of $\frac{\partial F(\gamma, y)}{\partial y}$ and the evolution of the filled regions in Figure 7. As the resulting dynamics becomes more restricted and less trivial, the following observations can be made: (1) the model tends to be less flexible and more difficult to fit the data, (2) the cost of the model is increasing with $\alpha_{div}$ and $\theta$, and (3) the explicit method demonstrates unstable oscillatory behavior when the level of dissipation exceeds its stability threshold.

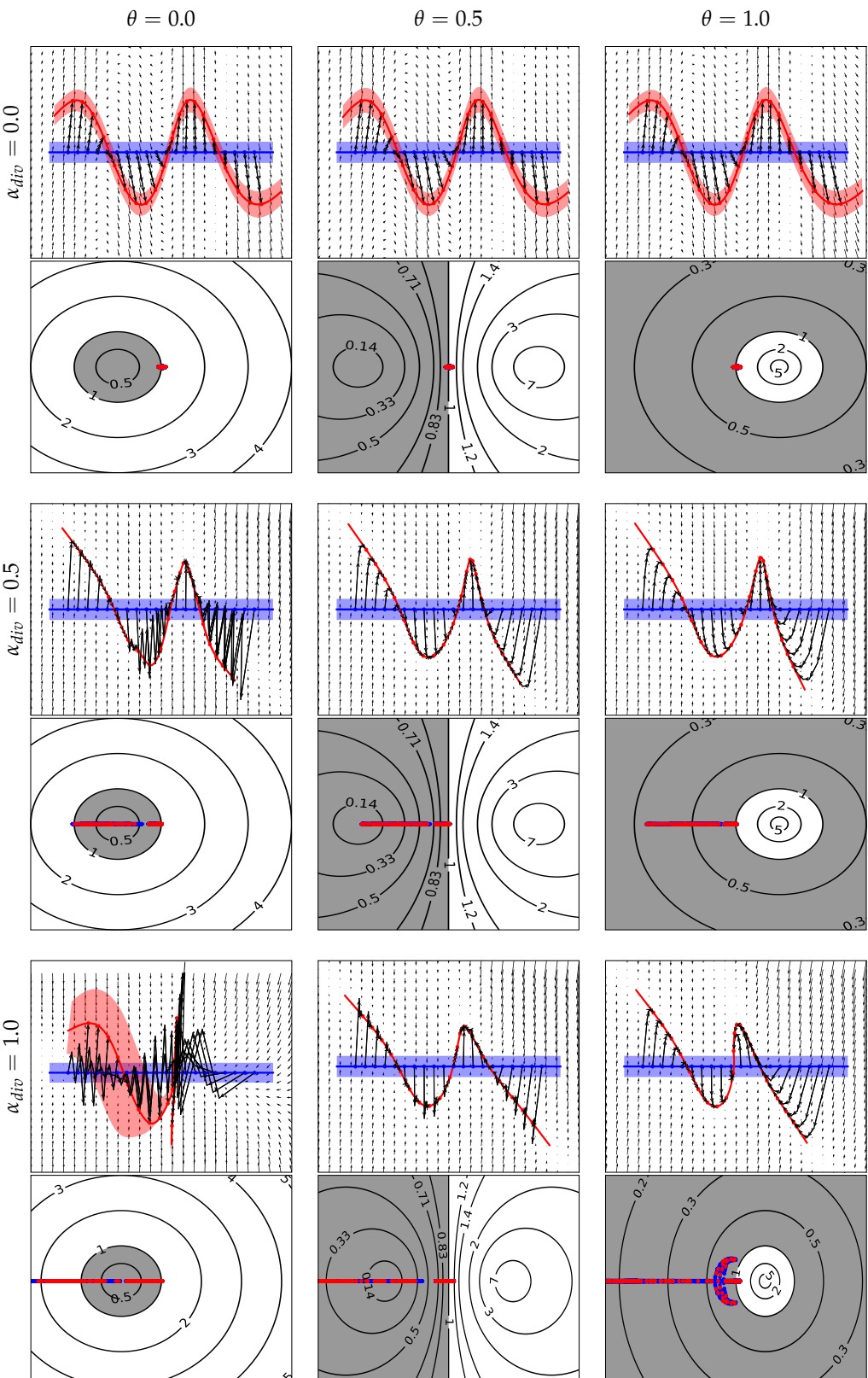

**Figure 7.** (**Top**) Learned vector fields $F(\gamma, y)$ in Example 1. Blue line is the initial state, red curve is the final state after $T = 5$ steps, solid black lines are the trajectories of the training data. (**Bottom**) Eigenvalues of $\frac{\partial F(\gamma, y)}{\partial y}$ evaluated along the learned trajectories at times $t = 0, \ldots, T$. Red and blue dots are used for the train and test datasets, respectively. Stability regions of implicit layers are highlighted with grey color and the contours depict the values of the stability function.

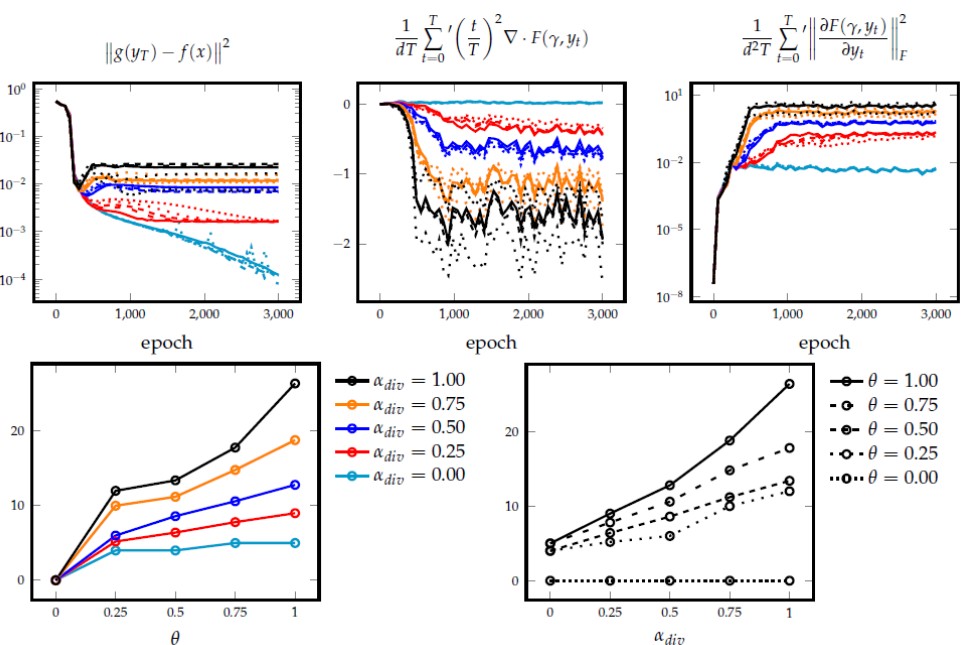

**Figure 8.** (**Top**) Evolution of the training loss components in (21) for Example 1. (**Bottom**) Nonlinear iterations per residual layer of the trained network.

### 3.2. Example 2. (Stiff ODE)

In our second example, we aim to fit the model to the given stiff ODE

$$\dot{z} = -20\big(z - \cos(t)\big), \quad t \in [0, 2]. \tag{22}$$

The training data in Figure 6 consist of 100 trajectories with randomly sampled initial conditions and the given number of uniformly distributed points along each trajectory as shown in Figure 9.

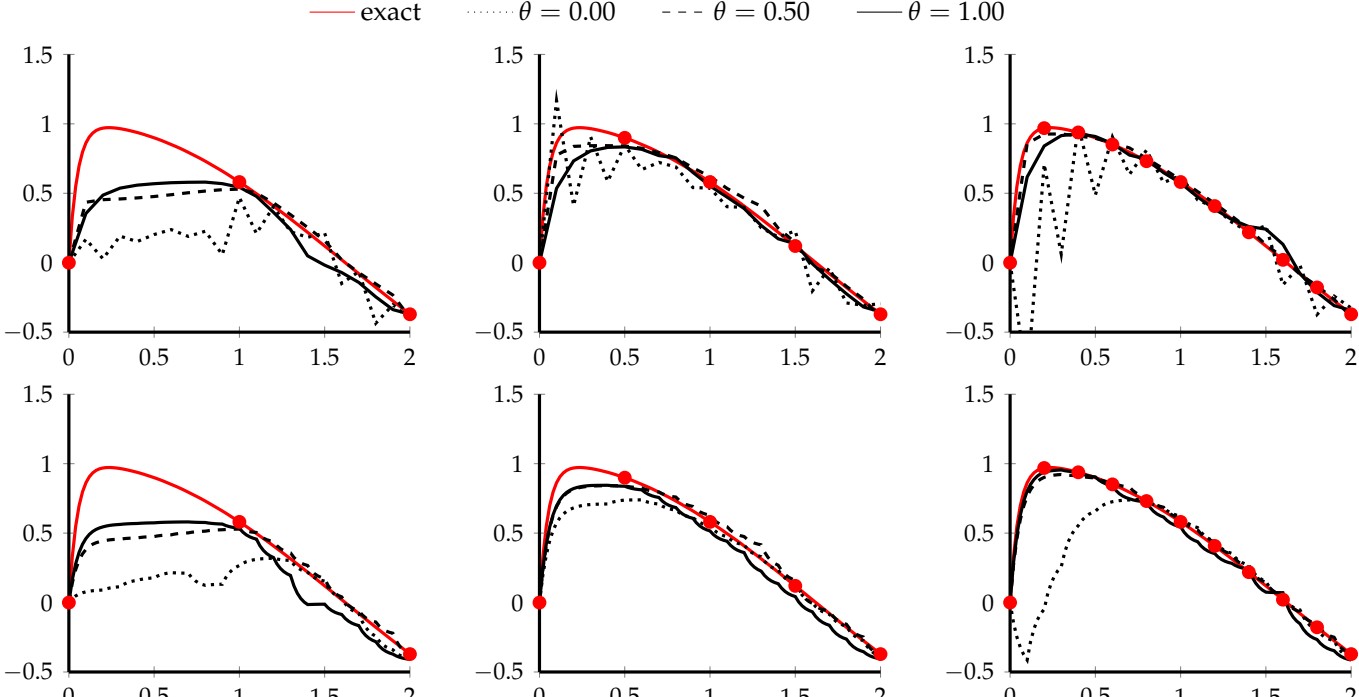

**Figure 9.** (Top) A single trajectory generated by three trained implicit residual networks for the problem in Example 2. (Bottom) Continuous-time trajectory generated by the learned vector fields of these residual networks.

Since the exact Lipschitz constant of the forcing term is equal to $-20$, we use $F^{-25,-15}(\gamma, x)$ in (19) with the one-dimensional vector field $F(\gamma, x) : \Gamma \times \mathbb{R}^1 \to \mathbb{R}^1$ given by the multilayer perceptron with 2 hidden layers of width 4 and ReLU activation function, i.e,

$$F(\gamma, y) = \gamma_{out}\sigma\left(\gamma_2\sigma\left(\gamma_1\sigma(\gamma_{in}y + b_0) + b_1\right) + b_2\right),$$

where $\gamma_{in} \in \mathbb{R}^{4\times1}$, $\gamma_{out} \in \mathbb{R}^{1\times4}$, $\gamma_i \in \mathbb{R}^{4\times4}$ and $b_i \in \mathbb{R}^{4\times1}$.

We took $T = 20$ residual layers initialized with Xavier uniform initializer and trained the network for 50 epochs using Adam optimizer, batch of size 1, and the regularized loss given by

$$L(\gamma) := \frac{1}{N \cdot \{2,4,10\}} \sum_{i=1}^{N} \sum_{j=1}^{\{2,4,10\}} \left\| y_j^i - z^i\left(t_0^i + \frac{j}{\{1,2,5\}}\right) \right\|^2 + \frac{0.1}{T} \sum_{t=1}^{T} \|\gamma_t - \gamma_{t-1}\|^2. \quad (23)$$

The top row of Figure 9 illustrates the evolution of the single trajectory generated by three residual models with $\theta = 0.0$, 0.5 and 1.0. Each model was trained using the loss function in (23) with 2, 4 and 10 points taken along the trajectories of the training dataset. One can see that the explicit ResNet is unstable as was expected for the stiff system in (22), and the generated solution becomes increasingly oscillatory along the transient part of the trajectory as we increase the number of the training points from 2 to 10. This is due to the model attempting to be increasingly expressive for the data it cannot potentially fit. Once the dynamical system relaxes to the slow manifold, the accuracy of the model improves slightly but remains susceptible to the orthogonal perturbations. In order to stabilize the model, more layers need to be taken which will lead to the increased memory footprint and computational complexity. At the same time, both implicit residual networks with $\theta = 0.5$ and 1.0 are unconditionally stable and improve their accuracy with increasing number of the training points as expected.

The bottom row of Figure 9 shows the continuous dynamics generated by the vector fields learned by three considered implicit models. In this case, the discrete-time stability is not an issue anymore. However, the corruption caused by the instability of the explicit method transfers to the inaccurate behavior of the continuous system as well. On the contrary, both implicit methods lead to the satisfactory approximation of the original vector field with the midpoint scheme ($\theta = 0.5$) being observably more accurate, likely due to its higher order of convergence. This suggests the proposed implicit networks as a means for learning stiff continuous-time ODE systems.

### 3.3. Example 3. (Periodic ODE)

In this example, we consider the Lotka–Volterra system given by

$$\begin{aligned}
\dot{z}_1 &= \alpha z_1 - \beta z_1 z_2, \\
\dot{z}_2 &= \delta z_1 z_2 - \gamma z_2
\end{aligned} \quad (24)$$

with $\alpha = \frac{2}{3}$, $\beta = \frac{4}{3}$, and $\delta = \gamma = 1$. These equations are used to model the time evolution of the biological systems with two interacting species one being the prey and the other being the predator. The model has two equilibrium points when neither of the two interacting populations is changing. The first equilibrium is at $z_1 = z_2 = 0$ and the non-trivial one is at $z_1 = \gamma/\delta = 1$, $z_2 = \alpha/\beta = \frac{1}{2}$. Other solutions are periodic and lie on the closed curves in the phase space. Figure 6 shows five such curves; each curve contains 51 points which are distributed uniformly on the time interval $t \in [0; 10]$ and used as the training data for our example.

To fit this data, we utilized implicit residual networks with $T = 50$ layers and the two-dimensional vector field approximated by the multilayer perceptron with four hidden layers of width 20 and ReLU activation functions, i.e.,

$$F(\gamma, y) = \gamma_{out}\sigma\left(\gamma_4\sigma\left(\gamma_3\sigma\left(\gamma_2\sigma\left(\gamma_1\sigma(\gamma_{in}y + b_0) + b_1\right) + b_2\right) + b_3\right) + b_4\right),$$

where $\gamma_{in} \in \mathbb{R}^{20\times2}$, $\gamma_{out} \in \mathbb{R}^{2\times20}$, $\gamma_i \in \mathbb{R}^{20\times20}$ and $b_i \in \mathbb{R}^{20\times1}$. We initialized the network with Xavier uniform initializer and trained it for 3000 epochs using Adam optimizer, full batch size, and the loss given by

$$L(\gamma) := \frac{1}{50 \cdot N} \sum_{i=1}^{N} \sum_{j=1}^{50} \left\| y_j^i - z^i(0.2j) \right\|^2.$$

Note that we did not use any form of weight normalization or regularization.

Figure 10 shows the learned trajectories and eigenvalues of the vector field along these trajectories for three implicit networks with $\theta = 0.0$, 0.5 and 1.0 on the time interval $t \in [0, 10]$. At a first glance, all three networks learn very similar vector fields and can accurately fit the data on the time interval of the training dataset. Moreover, the top row of Figure 11 shows that all three methods are also successful at extrapolating the dynamics to the time interval $t \in [0, 200]$. However, the bottom row of the same figure shows that the midpoint scheme ($\theta = 0.5$) is the only one which produces the vector field accurate for the long-time integration of the continuous dynamics. This behavior is indeed expected since the method is a geometrical integrator for this type of system, recall Figure 2 and the discussion in Section 2. The explicit residual network ($\theta = 0.0$) is strictly expanding for such conservative systems, and the learned vector field tends to compensate for this behavior resulting in the dissipative continuous dynamics. The situation is reverse for the backward Euler integrator ($\theta = 1.0$) since the method is strictly dissipative and hence the learned vector field is overly expanding.

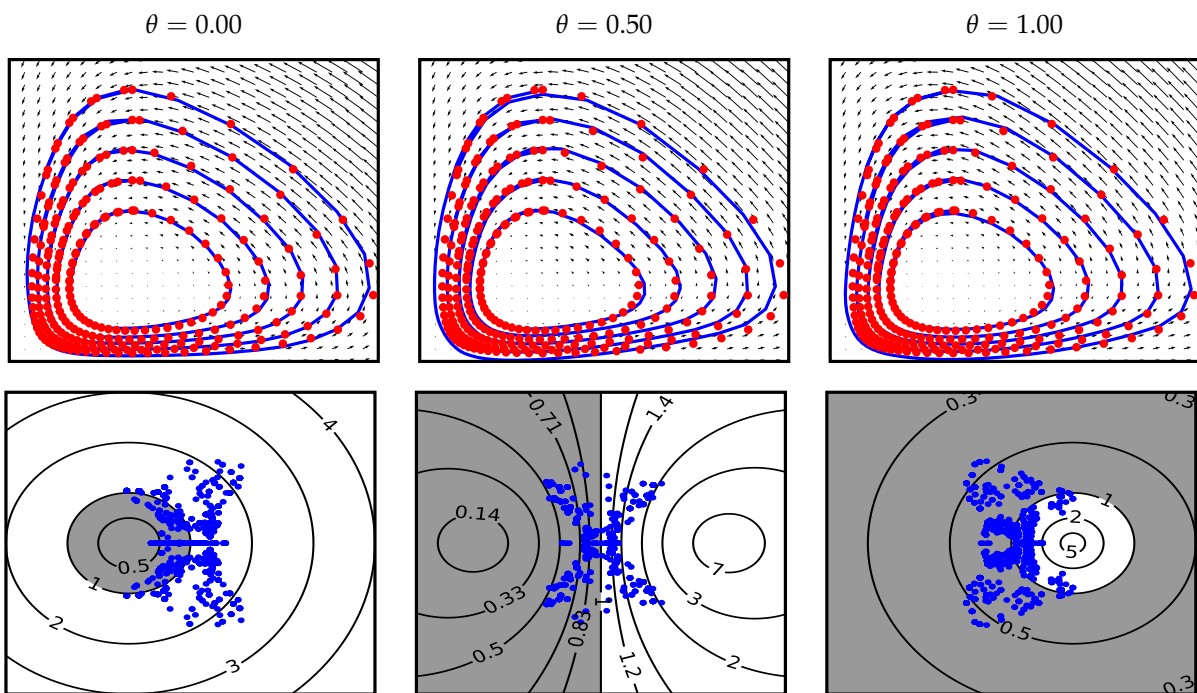

**Figure 10.** (**Top**) Learned vector fields and trajectories for the system in Example 3 on the time interval $t \in [0, 10]$. (**Bottom**) Eigenvalues of the vector fields along these trajectories.

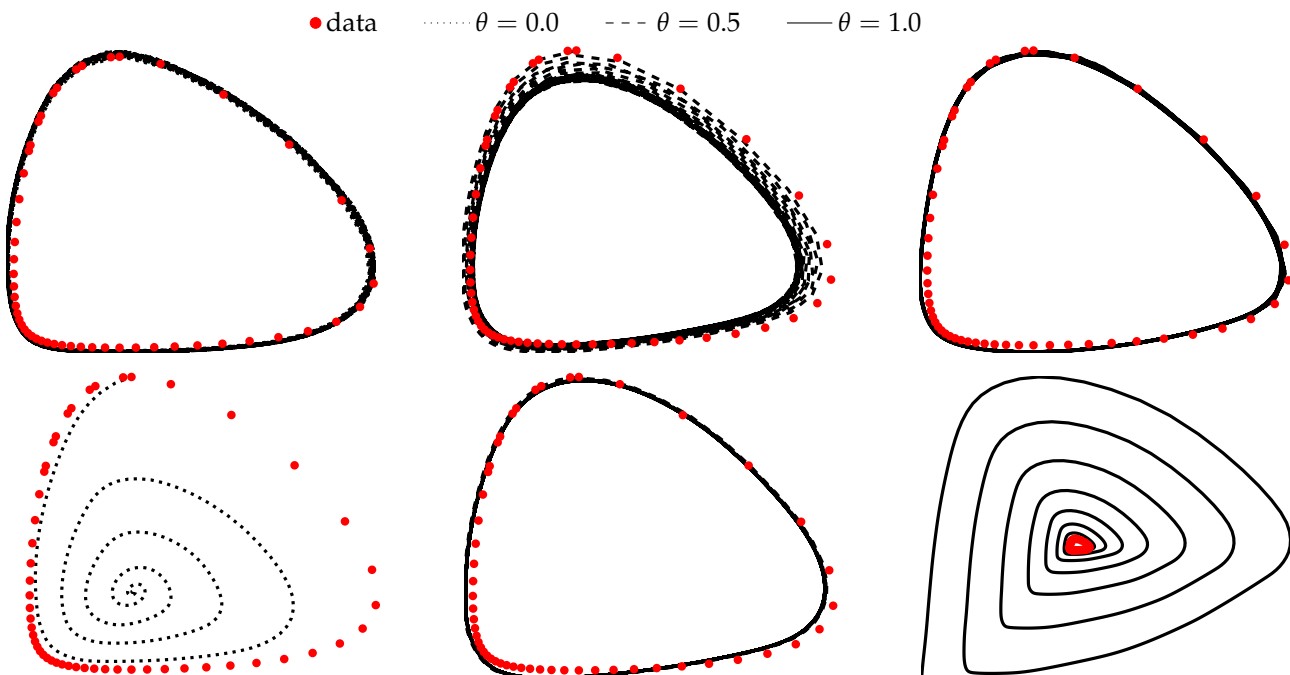

**Figure 11.** (**Top**) A single trajectory generated by three trained implicit residual networks for the problem in Example 3 on the time interval $t \in [0, 200]$; (**Bottom**) continuous-time trajectory generated by the learned vector fields of these residual networks on the same time interval.

### 3.4. Example 4. (MNIST Classification)

For the last example, consider the problem of classifying images of handwritten digits from the subset of MNIST dataset of size 1000. For this purpose, we adopt the standard preactivation ResNet-18 architecture with eight initial channels and train the network using the Adam optimizer for 100 epochs with learning rate of $10^{-2}$. We used the cross entropy loss function

$$L(\gamma) := \frac{1}{N} \sum_{i=1}^{N} \left[ -y_{label}^i + \log \left( \sum_{j=0}^{9} \exp(y_j^i) \right) \right]$$

and normalized the forcing terms $F(\gamma, x)$ of all residual layers as $F^{-3,1}(\gamma, x)$ using (19). Additionally, the divergence regularization in (20) with $\alpha_{div} = 0.01$ was applied to each residual layer.

To test the robustness of the trained network, we corrupted the original dataset with the Gaussian noise of varying standard deviation. Table 3 illustrates the classification accuracy for different levels of noise intensity and five different implicit residual networks. The results show that the implicit architectures with a proper regularization can significantly improve the robustness properties of trained networks.

**Table 3.** Classification accuracy in Example 4 for different levels of Gaussian noise corruption.

| Noise Intensity | Top-1 Accuracy | | | | | Top-2 Accuracy | | | | |
|---|---|---|---|---|---|---|---|---|---|---|
| | $\theta = 0$ | 0.25 | 0.50 | 0.75 | 1.00 | $\theta = 0$ | 0.25 | 0.50 | 0.75 | 1.00 |
| 0.0 | 100.0 | 100.0 | 100.0 | 100.0 | 100.0 | 100.0 | 100.0 | 100.0 | 100.0 | 100.0 |
| 0.1 | 98.2 | 99.6 | 99.9 | 99.9 | 99.9 | 99.7 | 100.0 | 100.0 | 100.0 | 100.0 |
| 0.2 | 89.2 | 95.7 | 96.3 | 98.3 | 98.4 | 96.7 | 99.3 | 99.8 | 100.0 | 100.0 |
| 0.3 | 74.7 | 86.0 | 89.3 | 93.2 | 93.6 | 89.0 | 95.2 | 97.8 | 99.0 | 98.8 |
| 0.4 | 59.3 | 74.1 | 77.2 | 81.8 | 84.7 | 75.5 | 89.1 | 91.1 | 94.4 | 95.0 |
| 0.5 | 47.2 | 60.4 | 64.7 | 69.8 | 73.0 | 65.7 | 79.6 | 83.4 | 87.1 | 87.9 |

## 4. Conclusions and Future Work

In this work, we presented a novel implicit residual layer and provided a memory-efficient algorithm to evaluate and train deep neural networks composed of such layers. We also proposed a regularization technique to control the spectral properties of the presented layer and showed that it leads to improved stability and robustness of the trained networks. The obtained numerical results support our findings.

We see several opportunities for potential improvements to the presented architecture. For example, implementations of the forward and backward propagation algorithms can be further optimized to account for the repetitive nature of the training process. This includes the better estimation of initial guesses for nonlinear solvers and preconditioners for linear solvers and other ways to reuse available information from previous runs. It is also interesting to study other, possibly multistep, types of implicit residual layers and their combinations. In this regard and in addition to the provided examples, we plan to identify the best use cases and applications for deep residual networks containing implicit layers. In particular, we are interested in exploring the impact of adversarial training and the proposed spectral regularization on the properties of the trained implicit networks. Other applications of interest include the identification of physical systems with known properties such as energy dissipation/conservation, large horizon time series forecasting, applications with corrupted and noisy data, etc. Finally, as the proper regularization is essential for the good performance of implicit layers, new regularization approaches tailored to specific applications should also be analyzed. We intend to study these questions in our future works.

**Author Contributions:** Conceptualization, V.R. and C.G.W.; methodology, V.R. and C.G.W.; software, V.R.; writing, V.R. and C.G.W.; funding acquisition, C.G.W. Both authors have read and agreed to the published version of the manuscript.

**Funding:** This research was funded by the U.S. Department of Energy, Office of Science, Early Career Research Program under award number ERKJ314; U.S. Department of Energy, Office of Advanced Scientific Computing Research under award numbers ERKJ331 and ERKJ345; the National Science Foundation, Division of Mathematical Sciences, Computational Mathematics program under contract number DMS1620280; and the Behavioral Reinforcement Learning Lab at Lirio LLC.

**Conflicts of Interest:** The authors declare no conflict of interest.

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
