# Peer review of "Robust Learning with Implicit Residual Networks"

_make, doi:10.3390/make3010003_

Round 1

Reviewer 1 Report

For implementation of deep neural networks(DNNs) authors propose an implicit residual architecture with a nonlinear transformation and they insist that the proposed method is unconditionally stable and robust.

The manuscript includes sufficient background and motivations of the main idea, and the authors' assertion is well-illustrated by the numerical results of the selected examples. 

As a result, I recommend publication of the manuscript if the conclusions are included appropriately.

Author Response

We would like to thank the referee for valuable comments and useful suggestions. We agree that the manuscript is incomplete without an appropriate conclusion. To address this comment, we added the following section to the revised manuscript:

Conclusions and future work.

In this work, we presented a novel implicit residual layer and provided a memory-efficient algorithm to evaluate and train deep neural networks composed of such layers. We also proposed a regularization technique to control the spectral properties of the presented layer and showed that it leads to improved stability and robustness of the trained networks. The obtained numerical results support our findings.

We see several opportunities for potential improvements to the presented architecture. For example, implementations of the forward and backward propagation algorithms can be further optimized to account for the repetitive nature of the training process. This includes the better estimation of initial guesses for nonlinear solvers and preconditioners for linear solvers and other ways to reuse available information from previous runs. It is also interesting to study other, possibly multistep, types of implicit residual layers and their combinations. In this regard and in addition to the provided examples, we plan to identify the best use cases and applications for deep residual networks containing implicit layers. In particular, we are interested to explore the impact of adversarial training and the proposed spectral regularization on the properties of the trained implicit networks. Other applications of interest include the identification of physical systems with known properties such as energy dissipation/conservation, large horizon time series forecasting, applications with corrupted and noisy data, etc. Finally, as the proper regularization is essential for the good performance of implicit layers, new regularization approaches tailored to specific applications should also be analyzed. We intend to study these questions in our future works.

Reviewer 2 Report

The present work proposes a new deep architecture utilizing residual blocks inspired by an implicit discretization scheme. The proposed reformulation of ResNet does not introduce new parameters and can potentially lead to a reduction in the number of required layers due to improved forward stability. Numerical results are presented to support the new findings.

Board Comments

Both the abstract and the introduction are well established. The methodology and the problem under analysis are identified. The manuscript is easy to read and follow, and finally, the results corroborate the statements. There are only two significant situations that would need clarification prior to publication in the reviewer point of view:

(1) The authors previously published, apparently, the same work in the 2nd Symposium on Machine Learning and Dynamical Systems (September 21 - 29, 2020). Nonetheless, this work is not cited. The authors should cite their previous work and clarify what breakthrough is reached with the present manuscript. Is it an extended work? What is extended?

(2) In the reviewer's opinion, there is excessive use of pre-prints to state the present work. Obviously, those pre-prints are valid, and even essential, to communicate ongoing work and work in a peer review stage. However, the reviewer does not understand when a pre-print has five years or more. About 25% of the references are pre-prints, most of them with more than two years.

Author Response

We would like to thank the referee for valuable comments and useful suggestions. A  point-by-point response to the comments is given below:

Comment: The authors previously published, apparently, the same work in the 2nd Symposium on Machine Learning and Dynamical Systems (September 21 - 29, 2020). Nonetheless, this work is not cited. The authors should cite their previous work and clarify what breakthrough is reached with the present manuscript. Is it an extended work? What is extended?

Response: The referee is correct that the work with the same title has been presented orally at the abovementioned symposium. It has not been published though. Moreover, we presented very preliminary results we had at that moment with a basic description of the network architecture and one numerical example. In the current manuscript, we provide a completely revised version of this work including an in-depth description of the method, a new regularization approach, and much extended numerical results. We have cited this presentation in the revision of the current manuscript and added the following paragraph at the end of the Introduction section:

The preliminary results for the work proposed in the current manuscript have been presented at the Second Symposium on Machine Learning and Dynamical Systems in the Fields Institute [25]. Here we provide a completely revised version of this work including an in-depth description of the method, a new regularization approach, and much extended numerical results.

Comment: In the reviewer's opinion, there is excessive use of pre-prints to state the present work. Obviously, those pre-prints are valid, and even essential, to communicate ongoing work and work in a peer review stage. However, the reviewer does not understand when a pre-print has five years or more. About 25% of the references are pre-prints, most of them with more than two years.

Response: We agree with the referee. In the revised manuscript, we have replaced pre-prints with the published versions for the following references:

2. Hanin, B. Universal function approximation by deep neural nets with bounded width and ReLU activations. Mathematics 2019, 7, 992. doi:10.3390/math7100992.

7. Ruthotto, L.; Haber, E. Deep Neural Networks Motivated by Partial Differential Equations. J. Math. Imaging Vis. 2020, 62, 352–364. doi:10.1007/s10851-019-00903-1.

18. Huang, G.; Liu, Z.; Van Der Maaten, L.; Weinberger, K.Q. Densely Connected Convolutional Networks. In Proceedings of 2017 IEEE Conference on Computer Vision and Pattern Recognition (CVPR), Honolulu, Hawaii, USA, 21-26 July 2017; IEEE Computer Society: Los Alamitos, CA, USA, 2017; pp. 2261–2269. doi:10.1109/CVPR.2017.243.

19. Haber, E.; Lensink, K.; Treister, E.; Ruthotto, L. IMEXnet A Forward Stable Deep Neural Network. In Proceedings of the 36th International Conference on Machine Learning (ICML 2019), Long Beach, California, USA, 9-15 June 2019; Chaudhuri, K.; Salakhutdinov, R., Eds.; PMLR, 2019; Vol. 97, pp. 2525–2534.

34. Miyato, T.; Kataoka, T.; Koyama, M.; Yoshida, Y. Spectral normalization for generative adversarial networks. In Proceedings of Sixth International Conference on Learning Representations (ICLR 2018), Vancouver, Canada, April 30 - May 3 2018; OpenReview.net, 2018.

35. Gouk, H.; Frank, E.; Pfahringer, B.; Cree, M. Regularisation of neural networks by enforcing Lipschitz continuity. Mach. Learn. 2020. doi:10.1007/s10994-020-05929-w.

39. Finlay, C.; Jacobsen, J.H.; Nurbekyan, L.; Oberman, A.M. How to train your neural ODE: the world of Jacobian and kinetic regularization. In Proceedings of the 37th International Conference on Machine Learning (ICML 2020), 12-18 July 2020; PMLR, 2020; Vol. 119, pp. 3154–3164. 

40. Kelly, J.; Bettencourt, J.; Johnson, M.J.; Duvenaud, D. Learning Differential Equations that are Easy to Solve. In Advances in Neural Information Processing Systems 33: 34st Annual Conference on Neural Information Processing Systems (NIPS 2020), 6-12 December 2020; Larochelle, H.; Ranzato, M.; Hadsell, R.; Balcan, M.F.; Lin, H.T., Eds.

42. Grathwohl, W.; Chen, R.T.; Bettencourt, J.; Sutskever, I.; Duvenaud, D. Ffjord: Free-form continuous dynamics for scalable reversible generative models. In Proceedings of Seventh International Conference on Learning Representations (ICLR 2019), New Orleans, Louisiana, USA, 6-9 May 2019; OpenReview.net, 2018.

To our knowledge, references [17, 32, 36] have not been published.

We have also attached the revised version of the references as the pdf file

Round 2

Reviewer 2 Report

The reviewer would like to thank the authors that have addressed accordingly all the concerns raised with the first review. It is the reviewer opinion that the manuscript is in optimal conditions to be published.